# Tumors attenuating the mitochondrial activity in T cells escape from PD-1 blockade therapy

Alok Kumar[†], Kenji Chamoto[†], Partha S Chowdhury, Tasuku Honjo*

Department of Immunology and Genomic Medicine, Graduate School of Medicine, Kyoto University, Kyoto, Japan

**Abstract** PD-1 blockade therapy has revolutionized cancer treatments. However, a substantial population of patients is unresponsive. To rescue unresponsive patients, the mechanism of unresponsiveness to PD-1 blockade therapy must be elucidated. Using a 'bilateral tumor model' where responsive and unresponsive tumors were inoculated into different sides of the mouse belly, we demonstrated that unresponsive tumors can be categorized into two groups: with and without systemic immunosuppressive property (SIP). The SIP-positive tumors released uncharacterized, non-proteinaceous small molecules that inhibited mitochondrial activation and T cell proliferation. By contrast, the SIP-negative B16 tumor escaped from immunity by losing MHC class I expression. Unresponsiveness of SIP-positive tumors was partially overcome by improving the mitochondrial function with a mitochondrial activator; this was not successful for B16, which employs immune ignorance. These results demonstrated that the 'bilateral tumor model' was useful for stratifying tumors to investigate the mechanism of unresponsiveness and develop a strategy for proper combination therapy.

*For correspondence:
honjo@mfour.med.kyoto-u.ac.jp

[†]These authors contributed equally to this work

Competing interests: The authors declare that no competing interests exist.

## Introduction

Cancer immunotherapy using immune checkpoint blockade, particularly antibodies against programmed cell death receptor 1 (PD-1) or its ligand (PD-L1), has made a revolution in cancer treatments as this treatment has durable response even to terminal stage cancers and lesser side-effects compared to the conventional cancer treatments (*Brahmer et al., 2010*; *Couzin-Frankel, 2013*; *Hodi et al., 2010*; *Mahoney et al., 2015*; *Topalian et al., 2015*). The success of clinical trials for the PD-1/PD-L1 axis blockade led the FDA to approve antibodies for PD-1 (e.g. nivolumab, pembrolizumab) or PD-L1 (e.g. Atezolizumab, Avelumab, Durvalumab) for different types of human cancers including metastatic non-small cell lung carcinoma (NSCLC), squamous cell lung cancer, renal cell carcinoma, hodgkin's lymphoma, head and neck squamous cell carcinoma, and recently, for microsatellite instability-high (MSI-H) or mismatch repair deficient (dMMR) cancers that include many late-stage cancers (*Chowdhury et al., 2018a*).

Despite the impressive success rate of PD-1 blockade therapy, a significant fraction of patients is unresponsive. To further improve its efficacy, we must (i) identify biomarker(s) that predict the responsiveness/unresponsiveness and (ii) develop improved strategy including the combination therapy. For these improvements, we need to understand the mechanism of unresponsiveness to PD-1 blockade therapy. Most studies on biomarkers and resistance mechanisms have focused only on the tumor's intrinsic properties (*Cristescu et al., 2018*; *Ribas, 2015*; *Rieth and Subramanian, 2018*; *Wellenstein and de Visser, 2018*; *Zou et al., 2016*). We need to elucidate the mechanism for unresponsiveness related to immune effector T cells to understand the complicated interaction between cancer and immunity. Several studies have worked on the unresponsive mechanism from the immunity side in different models. In one such model, the 'Cold and Hot tumor hypothesis', tumors can

**eLife digest** Immunotherapy is a fast-emerging treatment area that turns the body's own immune system against cancer. One powerful group of treatments are the PD-1 blockers. PD-1 is an inducible protein that is sometimes found on healthy immune cells called T cells and normally acts to stop T cells mistakenly attacking healthy cells. However, it can also prevent T cells attacking cancer. This happens when cancer cells make a protein called PD-1 ligand, which interacts with PD-1 to switch off nearby T cells. Antibodies that block PD-1 or PD-1 ligand can reactivate T cells, allowing them to destroy the cancer, but this PD-1 blocking therapy currently works in less than half of all patients who receive the treatment.

To mount a successful defense against cancer, a T cell needs to be able to perform two key tasks: recognize cancer cells and prepare to attack. T cells are alerted to the presence of the disease by MHC class I proteins on the surface of cancer cells holding up small fragments of molecules that are tell-tale sign that the cell is cancerous. To prepare to attack, a T cell depends on its mitochondria – the powerhouses of the cell – to send a cascade of signals inside the T cell that help it to activate and multiply. It is possible that cancer cells escape PD-1 blocking treatments by interfering with either one of these two tasks. They may either hide their MHC class I proteins to become invisible to passing T cells – a phenomenon known as "local immune ignorance"; or they may release long-range molecules to stop T cells preparing to attack – "systemic immune suppression".

To explore these options further, Kumar, Chamoto et al. developed a new tumor model in mice. Each mouse had two tumors, one that responded to PD-1 blocking treatment and one that did not. The idea was that, if the unresponsive tumor was simply hiding from passing T cells, its presence should not affect the other tumor. But, if it was releasing molecules to block T-cell activation, the other tumor could become unresponsive to PD-1 blocking treatment too.

Kumar, Chamoto et al. examined different types of unresponsive tumor in this model system and found that they fell into two groups. The first group simply hid themselves from nearby T cells, while the second group released molecules to dampen all T cells. The identity of these molecules is unknown, but further experiments suggested that they likely work by blocking the mitochondria in T cells. In mice with these tumors, drugs that boosted mitochondria activity made anti-PD-1 treatment more effective.

If the findings in this mouse model parallel those in humans, it could open a new research area for immunotherapy. The next step is for researchers need to identify the molecule responsible for systemic immune suppression. This could help to make PD-1 blocking treatments more effective in people who do not currently respond.

be roughly classified as 'immunologically hot (inflamed)' with an abundance of tumor-infiltrating lymphocytes (TILs) and 'immunologically cold (noninflamed)' with an absence of a sufficient population of pre-existing immune cells (*Bonaventura et al., 2019*; *van der Woude et al., 2017*). In addition, some groups claim that clinical failures in many patients could be due to an imbalance between T-cell reinvigoration and tumor burden (*Borcoman et al., 2018*; *Huang et al., 2017*).

CD8[+] T cells, the major immune effector cells for attacking tumors, are subject to negative regulation by multiple mechanisms in tumor-bearing hosts. Some of the well-known negative regulatory cells and soluble factors include myeloid-derived suppressor cells (MDSC), innate lymphoid cells (ILC), tumor-associated macrophages (TAM), regulatory CD4[+] T cells (Tregs), regulatory B cells (Bregs), transforming growth factor β (TGF-β), interleukin-10 (IL-10), adenosine, granulocyte-macrophage colony-stimulating factor (GM-CSF), prostaglandin E2 (PGE2), and L-Kynurenine (*Artis and Spits, 2015*; *DeNardo and Ruffell, 2019*; *Facciabene et al., 2012*; *Sarvaria et al., 2017*; *Tauriello et al., 2018*). Lack of MHC class I and neo-antigen on tumor cells also cause unresponsiveness because T cells cannot recognize the tumor (*Garrido et al., 2016*; *McGranahan et al., 2017*; *Rodríguez, 2017*). The tumor microenvironment, influenced by the above mechanisms, allows tumor cells to escape from immune attack (*DeNardo and Ruffell, 2019*; *Russo and Protti, 2017*). Due to this complexity of tumor and immunity interactions, it is difficult to determine which tumor employs which immune escape mechanism.

Energy metabolism mediated by mitochondrial activity regulates the fate of T cells. It has been reported that mitochondria play an important role in antigen-specific T cell activation through signaling of mitochondrial-derived reactive oxygen species (ROS) (*Mallilankaraman, 2018*; *Murphy and Siegel, 2013*; *Sena et al., 2013*). We recently reported that mitochondria are activated in tumor-reactive CTLs during PD-1 blockade therapy in MC38 tumor-bearing hosts (*Chamoto et al., 2017*). Boosting fatty acid oxidation with a metabolic modulator enhanced the PD-1 blockade effect (*Chowdhury et al., 2018b*). Therefore, attenuation of the mitochondrial activity of T cells by tumor-mediated factors could be an immune escape mechanism.

In this study, we developed a novel approach using a 'bilateral tumor model' and studied the immunosuppressive nature of unresponsive tumors to PD-1 blockade therapy. This model allowed us to categorize unresponsive tumors into two: those which have immune ignorance properties at tumor local sites and the others which have systemic immunosuppressive properties (SIP). SIP is mediated by small molecules to downregulate mitochondrial function directly and to inhibit T cell proliferation. Boosting the mitochondrial activity by the addition of bezafibrate, a pan-PPAR agonist, partially improved the efficacy of the PD-1 blockade against unresponsive tumors with SIP but not for tumors with immune ignorance at the local site.

## Results

### Different immune responses between hosts with responsive and unresponsive tumors

We first determined which tumor was responsive and unresponsive using an anti-PD-L1 monoclonal antibody (mAb) to block PD-1 signal (PD-1 blockade) or the *Pdcd1*[-/-] mouse model (*Figure 1—figure supplement 1*). As summarized in '*Table 1*', GL261, MC38, and MethA were characterized as responsive tumors while LLC, B16, Pan02, and CT26 were characterized as unresponsive tumors.

Since CD8[+] cytotoxic T lymphocytes (CTLs) are the main effector cells during PD-1 blockade therapy, we examined the difference in the host's immune responses to a responsive tumor and an unresponsive tumor according to the schedule shown in *Figure 1A*. We found both the total lymphocytes and the effector memory CD8[+] T cells (defined as CD62L[low] CD44[high], P3) in draining lymph nodes (DLNs) significantly increased in the group of responsive tumors, but did not change in unresponsive tumor-bearing hosts after PD-1 blockade (*Figure 1B and C*). Further, total CD44[+] T cells which include both central memory (CD62L[high] CD44[high], P2) and effector memory (CD62L[low] CD44[high], P3) were also larger after the PD-1 blockade therapy over ctrl IgG treated group in the hosts with responsive tumor (*Figure 1—figure supplement 2*).

**Table 1.** List of mouse cell lines from different genetic backgrounds used in this study.

| Cell line | Background | Response to PD-1 blockade therapy | Particulars | Source |
|---|---|---|---|---|
| GL261 | C57BL/6N | Responsive | Glioblastoma cell line | As a gift from Dr. Toda, Keio University, Japan |
| MC38 | C57BL/6N | Responsive | Colon carcinoma cell line | As a gift from Dr. James P. Allison, Memorial Sloan-Kettering Cancer Center (New York, NY, USA) |
| LLC | C57BL/6N | Unresponsive | Lewis lung carcinoma cell line | American Type Culture Collection |
| B16 | C57BL/6N | Unresponsive | Melanoma cell line | As a gift from Dr. Nagahiro Minato, Graduate School of Medicine, Kyoto University |
| Pan02 | C57BL/6N | Unresponsive | Pancreatic ductal adenocarcinoma cell line | National Cancer Institute |
| MethA | BALB/c | Responsive | 3-methylcholanthrene (MCA)-induced fibrosarcoma cell line | Cell Resource Center for Biomedical Research (Sendai, Japan) |
| CT26 | BALB/c | Unresponsive | N-nitroso-N-methylurethane-(NNMU) induced colon carcinoma cell line | National Cancer Institute |

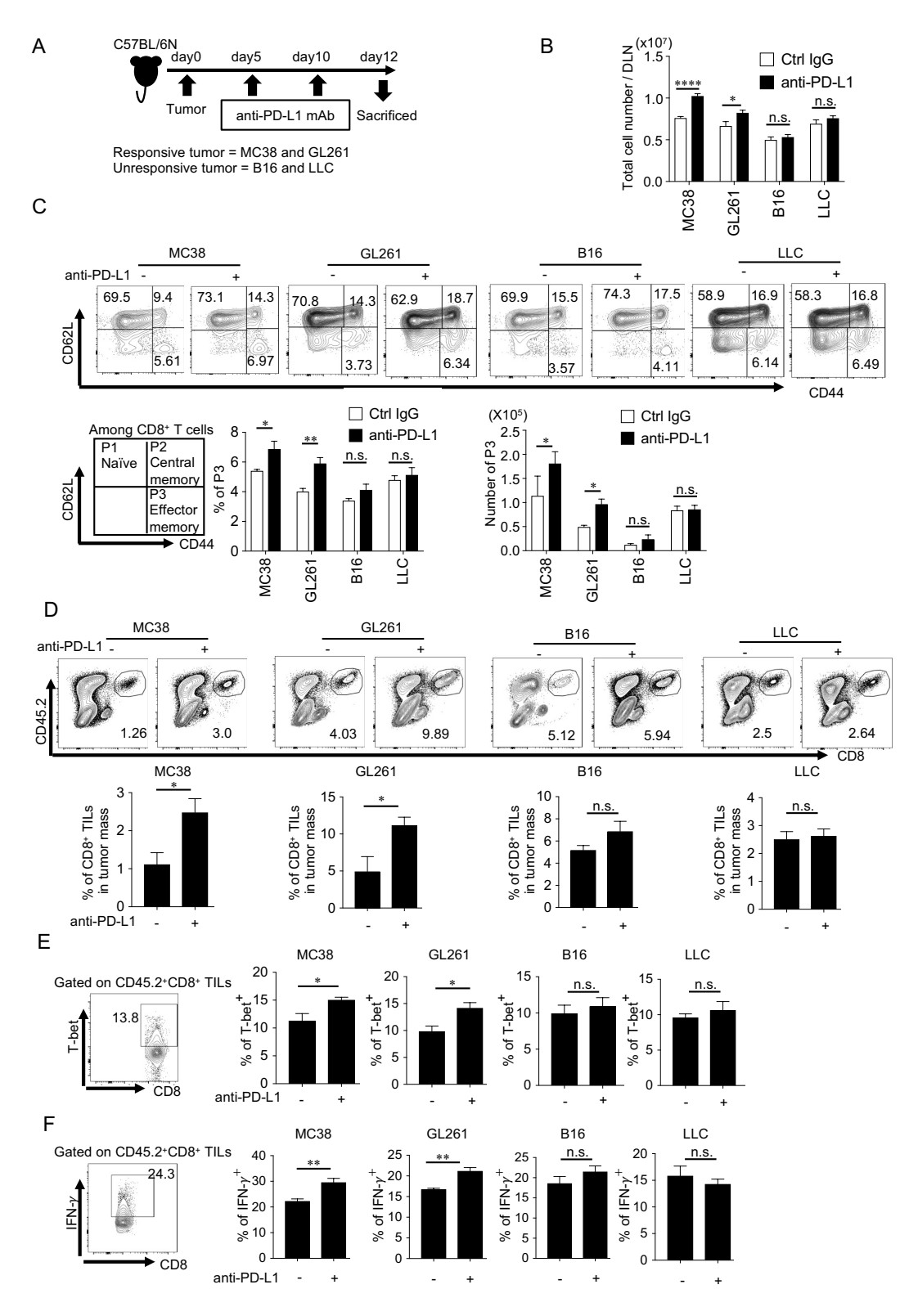

**Figure 1.** PD-1 blockade significantly enhances the number and function of effector CD8$^+$ T cells in mice with responsive, but not in those with unresponsive tumors. (**A**) Schematic diagram of the experimental schedule. (**B**) Absolute number of lymphocytes per draining lymph node (DLN) was calculated and compared among mice with different responsive or unresponsive tumors. (**C**) DLN cells were stained with anti-CD8, anti-CD62L, and anti-CD44 antibodies. Representative FACS patterns after gating on CD8$^+$ T cells in each group with or without PD-1 blockade (top panel). Schematic

*Figure 1 continued on next page*

Figure 1 continued

representation of subpopulations among CD8$^+$ T cells (bottom panel, left). Bar graphs of frequency and the absolute number of effector memory (CD62L$^{low}$ CD44$^{high}$; P3, hereinafter) population are shown (bottom panel, middle and right). (D) Cells after tumor digestion were stained with anti-CD8, and anti-CD45.2 antibodies. CD45.2$^+$ CD8$^+$ TIL frequency was compared between control IgG and anti-PD-L1 treated groups in responsive and unresponsive tumor-bearing hosts. Representative FACS pattern (upper panel) and the respective bar graph (lower panel) of CD45.2$^+$ CD8$^+$ TIL frequency are shown. (E) Harvested tumor mass cells from experimental groups were stained with anti-CD8, anti-CD45.2, and anti-T-bet antibodies. T-bet expression was plotted after gating on CD45.2$^+$ CD8$^+$ T cells. Representative FACS pattern from GL261 group (ctrl IgG treated) is shown (left). The frequency of T-bet among CD45.2$^+$ CD8$^+$ TILs of mice with the different tumor is shown. (F) IFN-γ expression was intracellularly analyzed in the same way as (E). Representative FACS pattern from GL261 group (ctrl IgG treated) is shown (left). The frequency of IFN-γ among CD45.2$^+$ CD8$^+$ TILs of mice with the different tumor is shown. (B–C) one-way ANOVA analysis. (D–F) two-tailed student's *t*-test analysis. *p<0.05, **p<0.01, ***p<0.001, ****p<0.0001, data represent the means ± SEM of five mice. Data are representative of two independent experiments. n.s. represents 'not significant'.

The online version of this article includes the following figure supplement(s) for figure 1:

**Figure supplement 1.** Stratification of responsive and unresponsive tumors in C57BL/6N and BALB/c genetic backgrounds.
**Figure supplement 2.** Increment of CD44$^+$ CD8$^+$ T cells in DLN after PD-1 blockade in the hosts with responsive tumor.
**Figure supplement 3.** Higher immune responses in responsive tumor-bearing host after PD-1 blockade compared to unresponsive group in BALB/c background.

The frequency of CD8$^+$ tumor-infiltrating lymphocytes (TILs) also increased after PD-1 blockade in the group of responsive tumor-bearing hosts, but not in unresponsive tumor-bearing hosts (*Figure 1D*). The expression of T-bet and IFN- γ, which reflect the activity of Th1-type cytotoxic activity, increased after PD-1 blockade treatment in the group bearing responsive tumors, but did not in the unresponsive tumor-bearing group (*Figure 1E and F*; *Sullivan et al., 2003*). Similar results were obtained in mice on another genetic background (BALB/c) (*Figure 1—figure supplement 3*). Taken together, anti-tumor immune responses increased only in hosts with responsive tumors but not in hosts with unresponsive tumors.

## Higher mitochondrial activity of effector CD8$^+$ T cells from mice with responsive tumors after PD-1 blockade

We and others have previously reported that mitochondrial activation in CD8$^+$ T cells is a marker of CTLs activation (*Buck et al., 2016*; *Chamoto et al., 2017*). Thus, to determine whether there was an association between the responsiveness to PD-1 blockade therapy and mitochondrial activation in T cells, we measured several markers of mitochondrial activation using the Seahorse Analyzer (*Figure 2—figure supplement 1A*). We found that DLN CD8$^+$ T cells from responsive (MC38 and GL261) tumor-bearing hosts had significantly higher basal respiration, maximal respiration, spare respiratory capacity (SRC), and ATP turnover by PD-1 blockade, which was not observed in unresponsive (B16 and LLC) tumor-bearing hosts (*Figure 2A*). Similar results were obtained in mice on the BALB/c background (*Figure 2—figure supplement 1B*). Besides, mitochondrial superoxide production (MitoSox) and Cellular ROS (CellRos) in CD8$^+$ TIL were increased by PD-1 blockade therapy only in responsive tumor-bearing mice (*Figure 2B and C*). Together, increased activity in CD8$^+$ T cells by PD-1 blockade in responsive tumor-bearing mice parallels with their activation status of mitochondria.

## Classification of unresponsive tumors by the presence or absence of systemic immunosuppressive property (SIP)

To investigate the mechanism of the systemic immune suppression of unresponsive tumors, we next employed a 'bilateral tumor inoculation model' where unresponsive and responsive tumors were inoculated on different sides of the host (*Figure 3A*). This model facilitates disclosing how much humoral factors derived from unresponsive tumors would contribute to the growth of responsive tumors in the other side. As shown in *Figure 3B*, we found that when unresponsive tumors (LLC or Pan02) were present on the left side, the growth inhibition of the responsive MC38 on the right by the PD-1 blockade therapy was inefficient. However, when the unresponsive B16 was on the left, the responsive MC38 or GL261 were rejected by PD-1 blockade as efficiently as the case in which no tumor was on the left side (*Figure 3B* and *Figure 3—figure supplement 1A*). The sizes of the left unresponsive tumor in the same experiment were not inhibited by the PD-1 blockade therapy

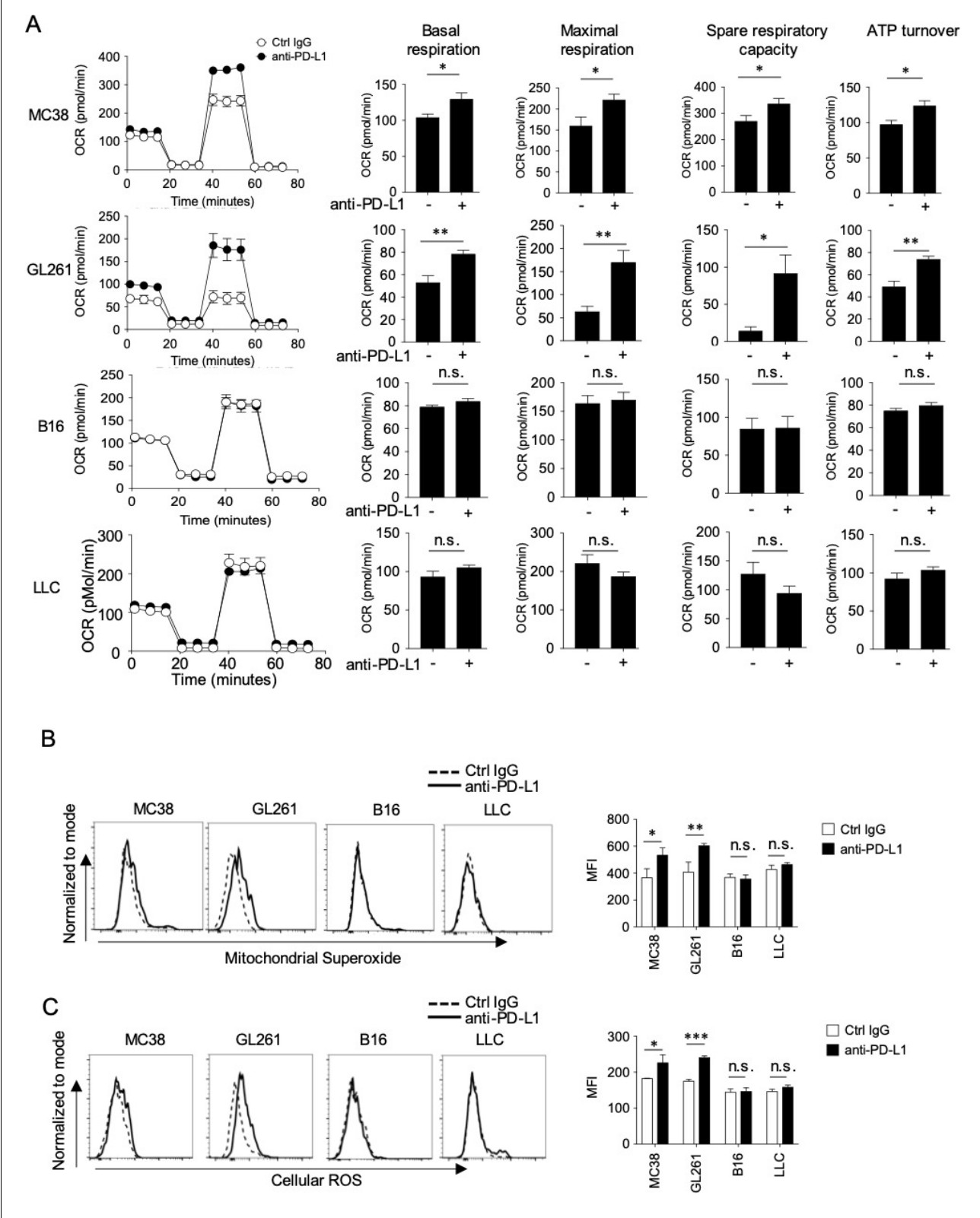

**Figure 2.** PD-1 blockade significantly enhances mitochondrial activity in CD8+ T cells in mice with responsive, but not in mice with unresponsive tumors. (A) DLN CD8+ T cells were purified from the pool of five mice per group from the experiment of *Figure 1*. OCR of DLN CD8+ T cells was measured from responsive and unresponsive tumor groups (left). Other parameters associated with OCR graph (basal respiration, maximal respiration, spare respiratory capacity, and ATP turnover) were calculated and values were plotted in a bar graph for respective tumor group (right). Data represent the

*Figure 2 continued on next page*

*Figure 2 continued*

means ± SEM of five wells. *p<0.05, **p<0.01, ***p<0.001, two-tailed student's *t*-test analysis. (**B–C**) Tumor mass cells, from the experimental groups of *Figure 1*, were stained with anti-CD8, anti-CD45.2 antibodies and mitochondrial dyes for Mitochondrial Superoxide production (**B**) or Cellular ROS production (**C**). Representative histogram (left) and MFI (right) of mitochondrial dyes after gating on CD45.2$^+$ CD8$^+$ T cells are shown. Data represent the means ± SEM of five mice. *p<0.05, **p<0.01, ***p<0.001, one-way ANOVA analysis. Data are representative of two independent experiments (**A–C**). n.s. represents 'not significant'.

The online version of this article includes the following figure supplement(s) for figure 2:

**Figure supplement 1.** CD8$^+$ T cells from mice with sensitive tumor have higher mitochondrial activity after PD-1 blockade than those with unresponsive tumors in BALB/c background.

(*Figure 3—figure supplement 1B*). Therefore, we speculated that the unresponsive LLC and Pan02 tumors may have released immune suppressive factors, while the unresponsive B16 did not.

Following the same experimental design, we performed the bilateral tumor experiment in mice on another background (BALB/c) and identified that CT26 is an unresponsive tumor with SIP (*Figure 3C* and *Figure 3—figure supplement 1C*). Taken together, we classified unresponsive tumors into two groups: those with or without SIP (*Table 2*).

### Tumor-derived suppressive soluble factor(s) systemically inhibits mitochondrial activity of CTLs in vivo

Since we observed mitochondrial activation in CD8$^+$ T cells as a parameter of responsiveness (*Figure 2*), we used the bilateral tumor model to investigate how immunosuppressive factors released from unresponsive tumors (on the left side) inhibited the immune response against responsive tumors (on the right side) from the aspect of mitochondrial activation (*Figure 4A*). As shown in *Figure 4B*, the absolute number of lymphocytes in the DLN on the side with MC38 was increased by PD-1 blockade in mice with the SIP-negative B16 on the other side, but not when the SIP-positive LLC was on the other side. Accordingly, mitochondrial ROS production, mass, OCR and ATP turnover in DLN CD8$^+$ T cells were also enhanced by PD-1 blockade on the MC38 side in the presence of B16 on the other side, but not the case when SIP-positive LLC was inoculated on the other side (*Figure 4C and D*). In contrast, the PD-1 blockade treatment did not change the mitochondrial activation status in the unresponsive tumor sides (B16 and LLC) (*Figure 4E and F*). In summary, while both LLC and B16 were unresponsive, only the LLC systemically inhibited the mitochondrial activation of CTLs during the PD-1 blockade therapy.

### The immunotherapy-resistant B16 tumor employs local immunological ignorance

We suspected that unresponsive tumors without SIP may not be recognized by the acquired immunity. We compared tumor growth between wild type and immune-compromised mice (Rag2$^{-/-}$). As shown in *Figure 5A*, the growth of responsive tumors (MC38, GL261, and MethA) was significantly restricted in wild type compared with Rag2$^{-/-}$ mice. In contrast, unresponsive tumors were more or less insensitive to acquired immunity (*Figure 5B*). Note that some unresponsive tumors with SIP (LLC and CT26) were sensed to a small extent by acquired immunity while unresponsive tumors without SIP (B16) were completely ignored (*Table 2*). This complete ignorance could be attributed to deficiencies in the 'T cell - tumor cell interaction' probably due to less neoantigen and/or lack of MHC class I expression. Indeed, we found that B16 does not express MHC class I even after stimulation with IFN-γ, but others do (*Figure 5C and D*). In other words, B16 acquired unresponsiveness by employing local immunological ignorance instead of SIP.

These data indicate that one of the mechanisms of unresponsiveness in tumors without SIP is lack of MHC class I expression, and suggest that elimination of the suppressive factor would facilitate the enhancement of PD-1 blockade therapeutic efficacy only in unresponsive tumors with SIP.

### Secretion of immune inhibitory small molecules from SIP-positive tumors

To examine whether immune suppressive factors are released from unresponsive tumors, naïve CD8$^+$ T cells were stimulated with anti-(CD3+CD28) mAb-coated beads in the presence of

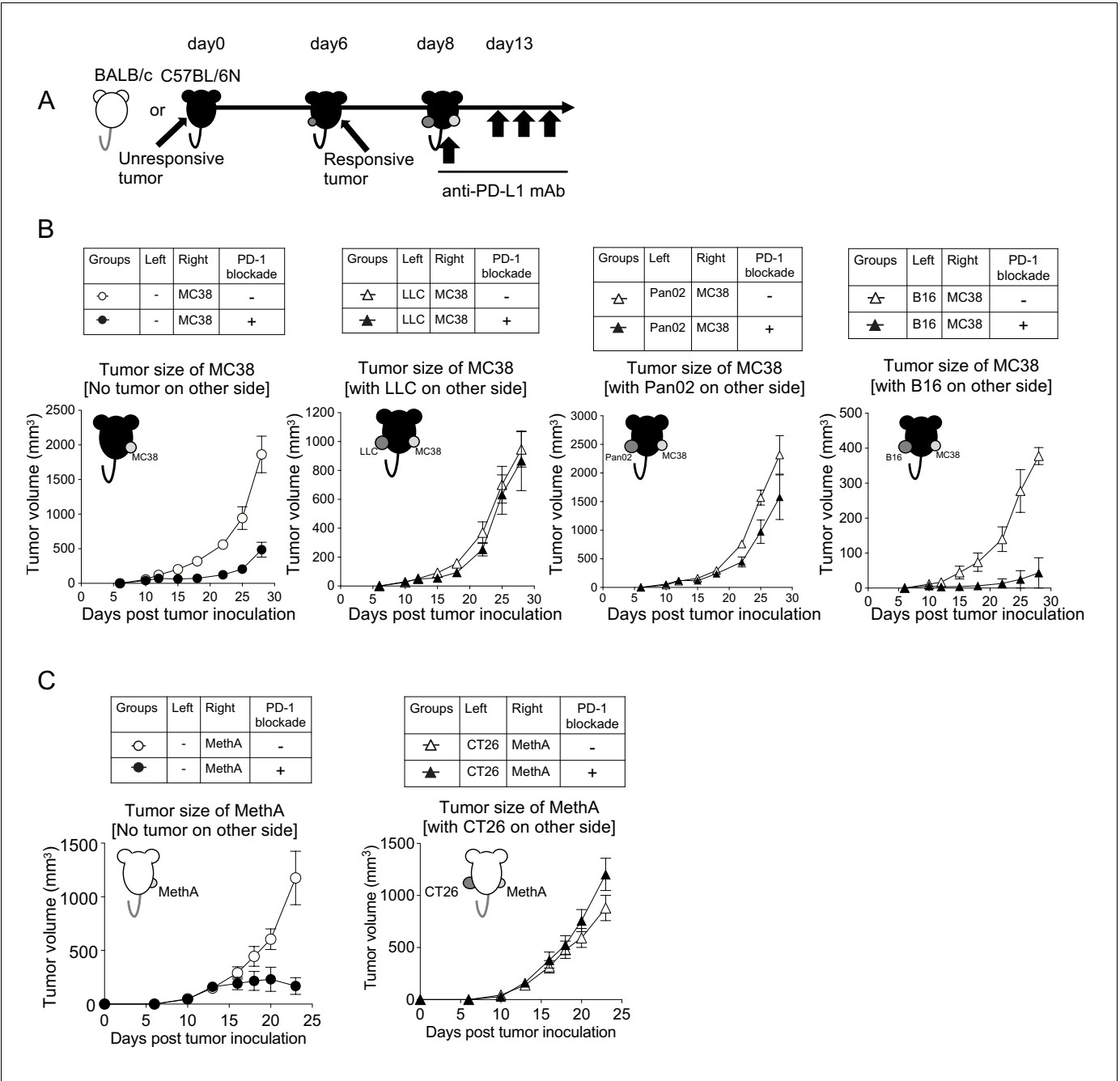

**Figure 3.** Unresponsive tumors can be classified into systemically immunosuppressive or non-immunosuppressive tumors. (A) Unresponsive tumor cells (LLC, Pan02 and B16) were inoculated on the left flank of C57BL/6N mice. On day 6, responsive tumor (MC38) cells were inoculated on the right flank of the same mice. On day 8, anti-PD-L1 mAb (or isotype control Rat IgG2a) was injected every fifth day thereafter. (B) Tumor growth of responsive MC38 on the right side was compared with or without PD-1 blockade treatment. (C) Following the same schedule, as mentioned in (A), unresponsive tumor (CT26) cells and responsive tumor (MethA) cells were injected in BALB/c mice. Tumor growth of MethA on the right side was shown. (B–C) Data represent the means ± SEM of five mice. Data are representative of two independent experiments.

The online version of this article includes the following figure supplement(s) for figure 3:

**Figure supplement 1.** Unresponsive tumors can be classified into systemically immunosuppressive or non-immunosuppressive tumors.

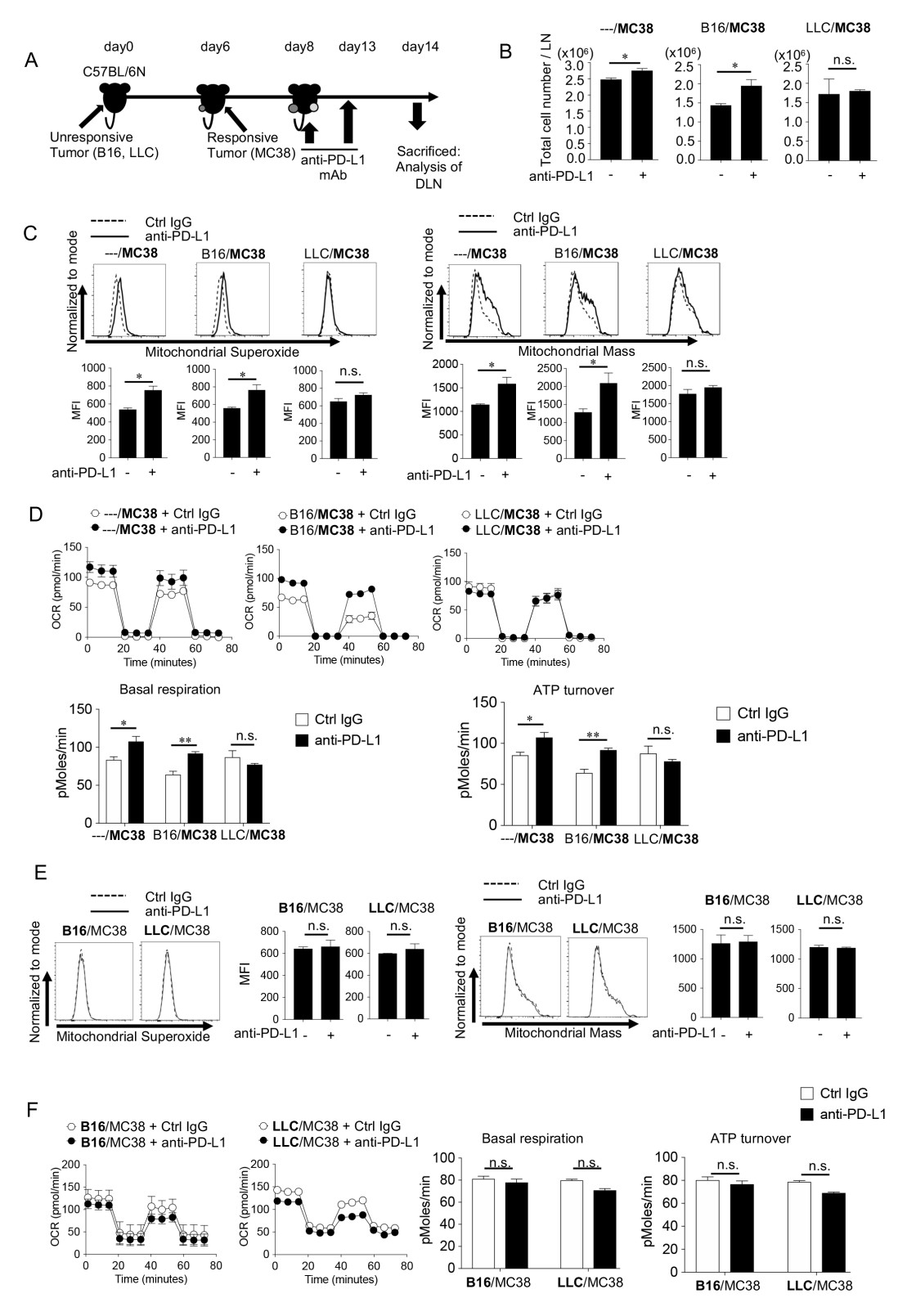

**Figure 4.** Unresponsive tumor-derived immune suppressive factor inhibits the mitochondrial responses in CD8[+] T cells in vivo. (**A**) Mice were treated in the same way as *Figure 3A* and sacrificed on day 14 for the analysis of DLN CD8[+] T cells. (**B**) Absolute number of lymphocytes per LN from the MC38 side was calculated. (**C**) DLN cells harvested from the MC38 side were stained with anti-CD8 mAb, MitoSox (left panels) and MitoMass (right panels). Representative FACS profiles after gating on CD8[+] T cells and MFI of dye staining are shown. (**D**) OCR of CD8[+] T cells purified from pooled DLN cells

*Figure 4 continued on next page*

Figure 4 continued

of MC38 side for different groups is shown (top). Basal respiration and ATP turnover values were calculated from the OCR graph (bottom). (**E**) DLN cells harvested from the unresponsive side were stained with anti-CD8 mAb, MitoSox (left) and MitoMass (right). Representative FACS profiles of DLN CD8$^+$ T cells and the MFI of dye staining are shown. (**F**) OCR of CD8$^+$ T cells purified from pooled DLN cells of B16 or LLC side is shown (left). Basal respiration and ATP turnover values were calculated from the OCR graph (right). (**B, C, E**) Data represent the means ± SEM of five mice. *p<0.05, **p<0.01, two-tailed student's *t*-test analysis. (**D, F**) Data represent the means ± SEM of five wells. *p<0.05, **p<0.01, one-way ANOVA analysis. Data are representative of two independent experiments. n.s. represents 'not significant'.

supernatants collected from responsive and unresponsive tumor cell cultures (*Figure 6A*). Proliferation assays (thymidine incorporation and Ki67 detection assays) demonstrated that T cell proliferation was significantly inhibited in the presence of supernatants from LLC or CT26, but not in the presence of supernatants from B16, GL261 or MethA (*Figure 6B* and *Figure 6—figure supplement 1A and B*). The suppressive effects of soluble factors from the LLC supernatant was further evidenced by the restoration of T cell proliferation when the supernatant was diluted (*Figure 6—figure supplement 1C*). It is of note that the SIP factor production is not only specific to mouse cell lines, but also to human cell lines (*Figure 6—figure supplement 2*).

In addition, different parameters of mitochondrial activation such as cellular ROS and mitochondrial potential were significantly inhibited by the LLC supernatant compared with the B16 and GL261 supernatants (*Figure 6C*). The OCR and the extracellular acidification rate (ECAR), a parameter for glycolytic function, were severely reduced in CD8$^+$ T cells cultured for 48 hrs in the presence of LLC supernatants compared with those from B16 and GL261 (*Figure 6D and E*). Similar suppressive activities were observed by supernatants from BALB/c background tumor CT26 (*Figure 6—figure supplement 1D*). To clarify whether this mitochondrial suppression is direct or bystander, we examined mitochondrial activation parameters within 2 hrs of coculture with the supernatant. As shown in *Figure 6F*, mitochondrial activation parameters were inhibited in the presence of LLC supernatants immediately, indicating that SIP factors highly likely inhibit mitochondrial activity directly, but not cellular transcriptional activity. Indeed, the transcriptional levels of PGC-1α/β, a master regulator of mitochondrial activation, did not change within 2 hrs (*Figure 6—figure supplement 3A*). The SIP factor inhibited B cell mitochondria as well within 2 hrs, showing this suppressive effect is more general (*Figure 6—figure supplement 3B*). These results indicate that the immunosuppressive factors released from SIP-positive tumors directly and generally inhibit the mitochondrial function.

Further, to understand the molecular properties of suppressive factors, we performed heat-inactivation to denature protein components and used a dextran-coated charcoal (DCC) treatment to adsorb small molecules in the culture supernatants. As shown in *Figure 6G* and *Figure 6—figure supplement 1E*, heat-inactivation of LLC and CT26 culture supernatants did not abolish their suppressive activity, whereas removing low molecular weight compounds using the DCC treatment eliminated their suppressive activity, suggesting that the suppressive factor(s) may be comprised of nonproteinaceous small molecules. We further fractionated the supernatant into 'Fraction A (<3 KDa)' and 'Fraction B (3 ~ 50 KDa)' and found that 'Fraction A' had almost the same inhibition potential as the total culture supernatants (*Figure 6H* and *Figure 6—figure supplement 1F*). Again, removing small molecules from 'Fraction A' using the DCC treatment restored the proliferation of CD8$^+$ T cells. We further tested whether previously reported small molecules could be candidates of the SIP factor such as adenosine, Prostaglandin E2 (PGE$_2$) and kynurenine, the transcriptional levels of key enzymes to produce them were examined. However, there was no relationship between the

**Table 2.** Mechanistic classification of unresponsive tumors.

| Background | Name of tumor | Releasing suppressive factor (related to *Figure 3*) | Activation of acquired immunity (related to *Figure 5*) |
|---|---|---|---|
| C57BL/6N | B16 | No | No |
|  | LLC | Yes | Less |
|  | Pan02 | Yes | No |
| BALB/c | CT26 | Yes | Less |

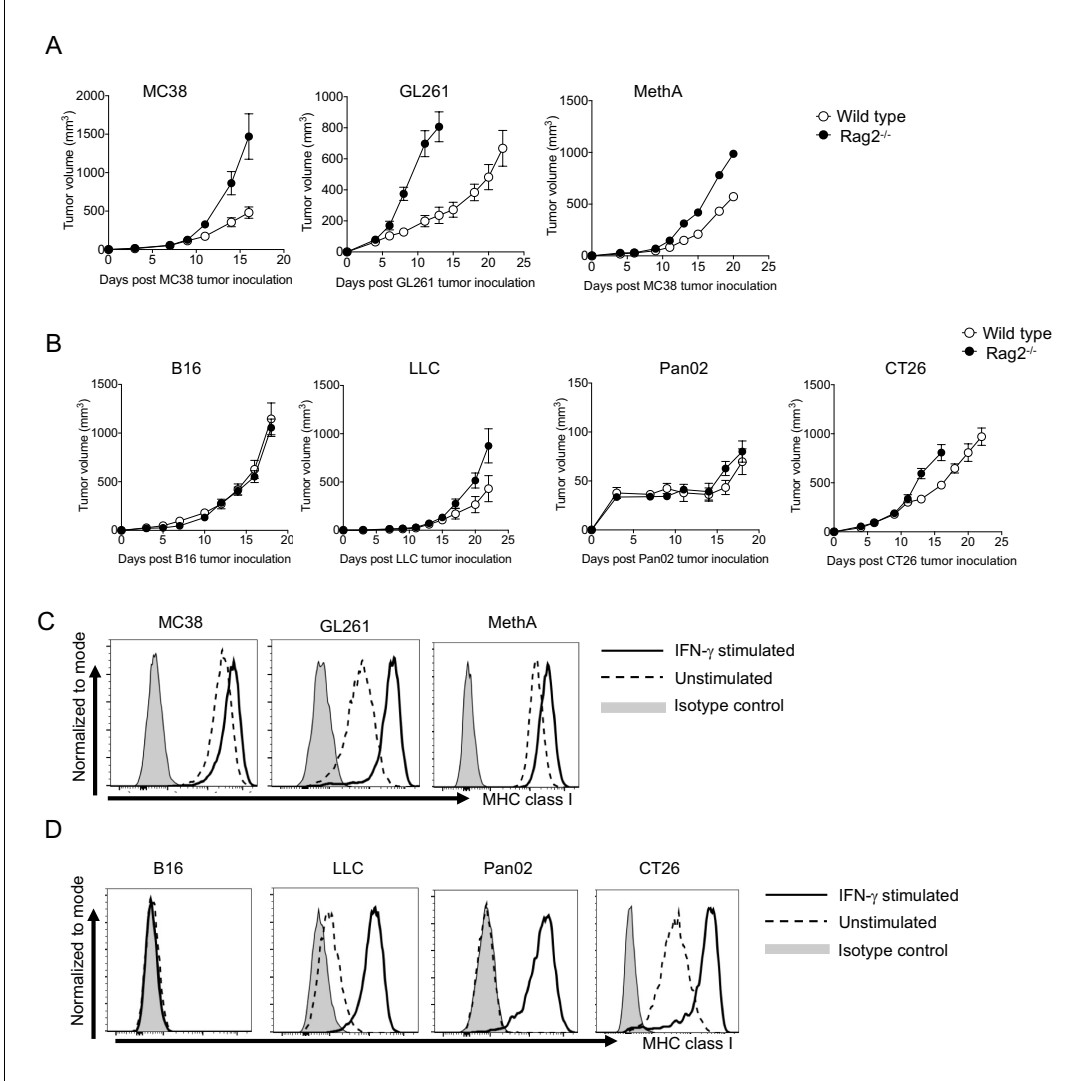

**Figure 5.** The absence of MHC class I expression in B16. (A–B) Tumor growth of responsive and unresponsive tumors was observed in wild type or immune-compromised (Rag2$^{-/-}$) mice. Tumor sizes of responsive tumors (A) and unresponsive tumor (B) are shown. Data represent the means ± SEM of 5 mice. (C–D) Responsive and unresponsive tumor cells were stimulated with IFN-γ for overnight, followed by staining with anti-H-2Kb/H-2Kd mAbs. Representative histograms of MHC class I for responsive (C) and unresponsive (D) tumor cells are shown. Data represent the means ± SEM of three wells. Data are representative of three independent experiments.

suppressive property and the expression levels of enzymes including CD39, CD73, COX-2, mPGES1 and IDO1 (*Figure 6—figure supplement 4*), suggesting the low possibility of known factors.

## Combination of bezafibrate with PD-1 blockade improves survival of mice bearing SIP-positive tumors

Since SIP reduced the mitochondrial activity, we examined whether mitochondria activation drug combination can reverse the immune suppression by SIP-positive tumors. As bezafibrate activates mitochondria and synergizes with PD-1 blockade therapy, we first tested whether bezafibrate can reverse the suppression of mitochondrial function and proliferation caused by suppressive factors from the LLC culture supernatants in vitro (*Chowdhury et al., 2018b*). Mitochondrial function of naïve CD8$^{+}$ T cells was regained significantly when bezafibrate was used along with culture supernatant in vitro (*Figure 7A*). Encouraged with these in vitro results, we performed PD-1 blockade combinatorial therapy with bezafibrate for LLC tumor-bearing hosts (*Figure 7B*). We found that the tumor-killing effect by the PD-1 blockade was enhanced and mouse survival was increased in the

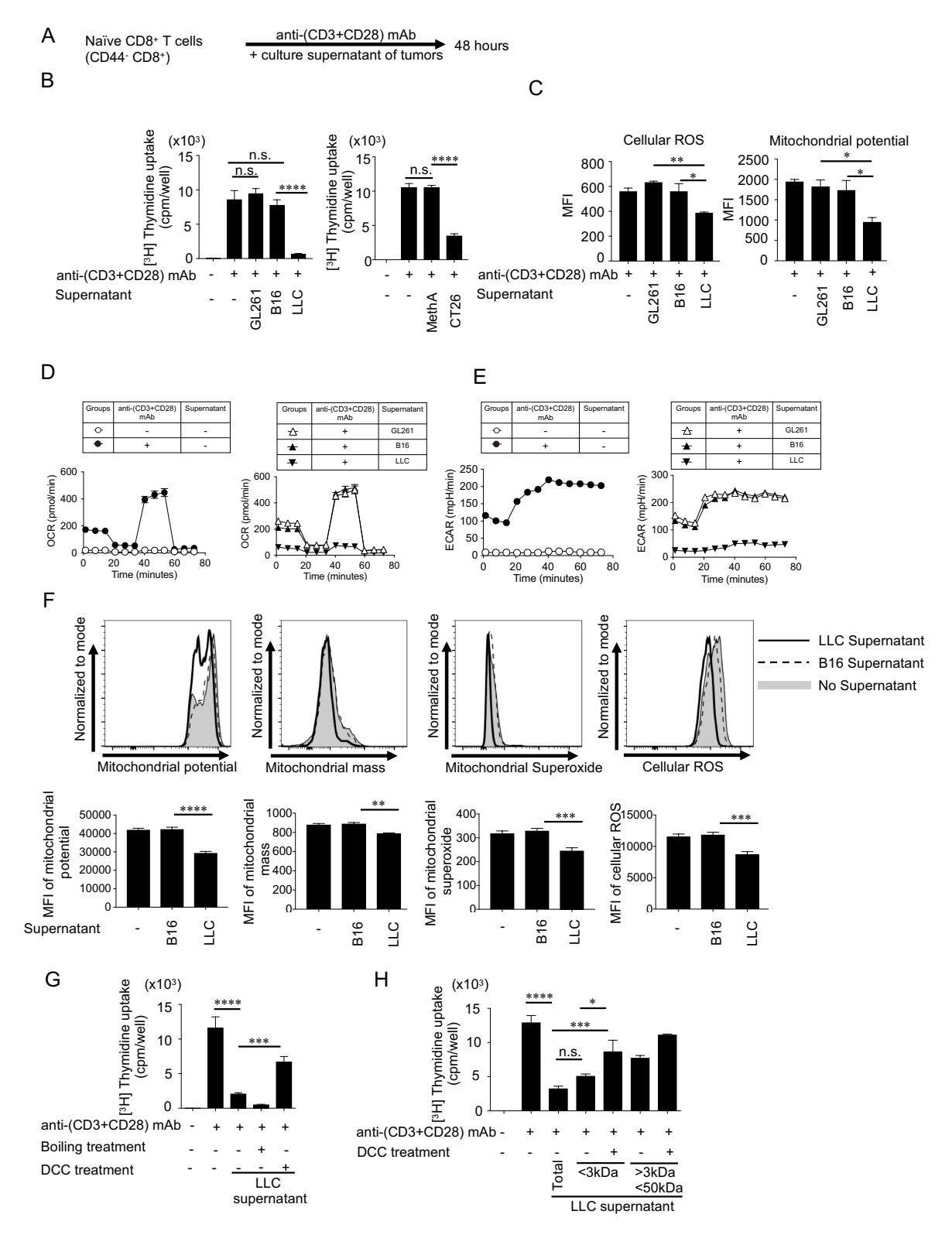

**Figure 6.** Small soluble factors released from SIP-positive tumors inhibit the T cell proliferation and mitochondrial function in vitro. (**A**) Naïve CD8[+] T cells (CD44[-] CD8[+] T cells) were purified from spleen and LNs of C57BL/6N mice. Naïve CD8[+] T cells were stimulated with anti-(CD3+CD28) mAbs-coated dynabeads for 48 hrs with or without culture supernatant from different tumor cell lines. (**B**) T-cell proliferation was measured by [3]H-thymidine incorporation assay. (**C**) T cells were stained with anti-CD8 mAb, CellRox dye (cellular ROS, left) and MitoTracker Deep Red dye (mitochondrial

*Figure 6 continued on next page*

*Figure 6 continued*

potential, right) after the stimulation. The MFI of mitochondrial dyes of CD8$^+$ T cells are shown. (D–E) OCR (D) and ECAR (E) of T cells were measured. The OCR graphs without (left) or with culture supernatants groups (right) are shown. (F) Naïve CD8$^+$ T cells were stimulated with B16 or LLC supernatant for 2 hrs. In the control wells, fresh medium was added. Following the stimulation, cells were stained with mitochondrial dyes. Representative histogram (upper panel) and MFI (lower panel) of each dye are shown. (G) LLC supernatant was heat-inactivated to denature protein components. To remove small molecules, the supernatant was treated with dextran-coated charcoal (DCC) that adsorbs small molecules. The effects of treated supernatant on T cell proliferation was assessed. (H) Using different cut-off filters, LLC supernatant was fractionated into <3 kDa and <50 kDa fractions that were further treated with DCC. The effects of the treated fractions on naïve CD8$^+$ T cell proliferation was assessed. Data represent the means ± SEM of three wells. *p<0.05, **p<0.01, ***p<0.001, ****p<0.0001, one-way ANOVA analysis (B–H). Data are representative of three independent experiments. n.s. represents 'not significant'.

The online version of this article includes the following figure supplement(s) for figure 6:

**Figure supplement 1.** Tumor-derived suppressive factor inhibits proliferation and mitochondrial function of CD8$^+$ T cells in vitro.

**Figure supplement 2.** Some human cancer cell lines release suppressive factors which inhibit T cells proliferation in vitro.

**Figure supplement 3.** Soluble suppressive factors inhibit mitochondria.

**Figure supplement 4.** Previously reported factors are not related with the nature of SIP- positive tumors.

combination therapy (*Figure 7C*). Of note is the fact that the combinatorial treatment could not rescue the B16 tumor-bearing hosts (*Figure 7C*). We observed similar results in tumors on the BALB/c background. The survival of SIP-positive CT26 tumor-bearing hosts was improved with the combinatorial therapy with bezafibrate (*Figure 7—figure supplement 1*). In summary, the SIP effects of unresponsive tumors were partially rescued by a mitochondrial activation chemical, bezafibrate in vitro and in vivo.

## Discussion

One of the biggest issues in PD-1 blockade cancer immunotherapy is how to reduce the rate of unresponsiveness. Although there are many unresponsive mechanisms, cancers employ at least two strategies to escape from the immune attack: local or systemic immune suppression. Some reports have suggested 'hot tumors' and 'cold tumors' to distinguish responsive and unresponsive tumors based on the level of immune cell infiltration in the tumor mass (*van der Woude et al., 2017*). However, it is difficult to explain the molecular mechanisms of unresponsiveness by this definition because it explains the results of immune responses in local tumor areas, but not the induction phase of immune escape.

In this paper, we employed the bilateral tumor inoculation model, which can distinguish local immune ignorance from systemic immune suppression, and categorized unresponsive tumors into two groups, with or without SIP. Small molecule(s) with less than 3 kDa size which is released from SIP-positive tumors appear to attenuate mitochondria-mediated energy metabolism in T cells. We rule out the known factors such as suppressive cytokines, adenosine, Prostaglandin E2 (PGE$_2$) and kynurenine. Tumor cells show dysregulated cellular metabolism and the metabolic products often induce immune suppression (*DeBerardinis, 2008*; *Munn and Mellor, 2013*; *Vazquez et al., 2016*). Although it has been reported that methyl-nicotinamide (MNA), which is converted by nicotinamide N-methyl-transferase (NNMT), acts as an immune suppressive factor (*Gebicki et al., 2003*), this compound showed no suppression at physiological levels (data not shown). Other metabolites could be candidates, which are derived from the tumor's metabolic activity.

For successful PD-1 blockade therapy, the 'tumor-immunity cycle' needs to operate smoothly (*Chen and Mellman, 2013*; *Pio et al., 2019*). Hindrance in the pathway at any step of antigen recognition, activation, recruitment and killing at the tumor site, DLN or bloodstream would lead to the unresponsive state (*Mushtaq et al., 2018*). DLN is generally considered as a place where naïve T cells are primed to effector T cells. Our bilateral tumor model analysis suggests that LLC systemically inhibits T cell priming at DLN of responsive tumor sides via suppressive factors, but B16 does not. However, it seems to contradict that T cells in DLN on the side of B16 were not activated in spite of the deficiency of SIP. This observation suggests that tumor recognition by the local tumor area is critical to trigger T cell priming in DLN and to establish a successful tumor-immunity cycle. Therefore, tumors lacking MHC take advantage of the ignorance or escape mechanism not only in the local tumor area but also in DLN. Given that LLC expresses MHC and is sensitive to the acquired

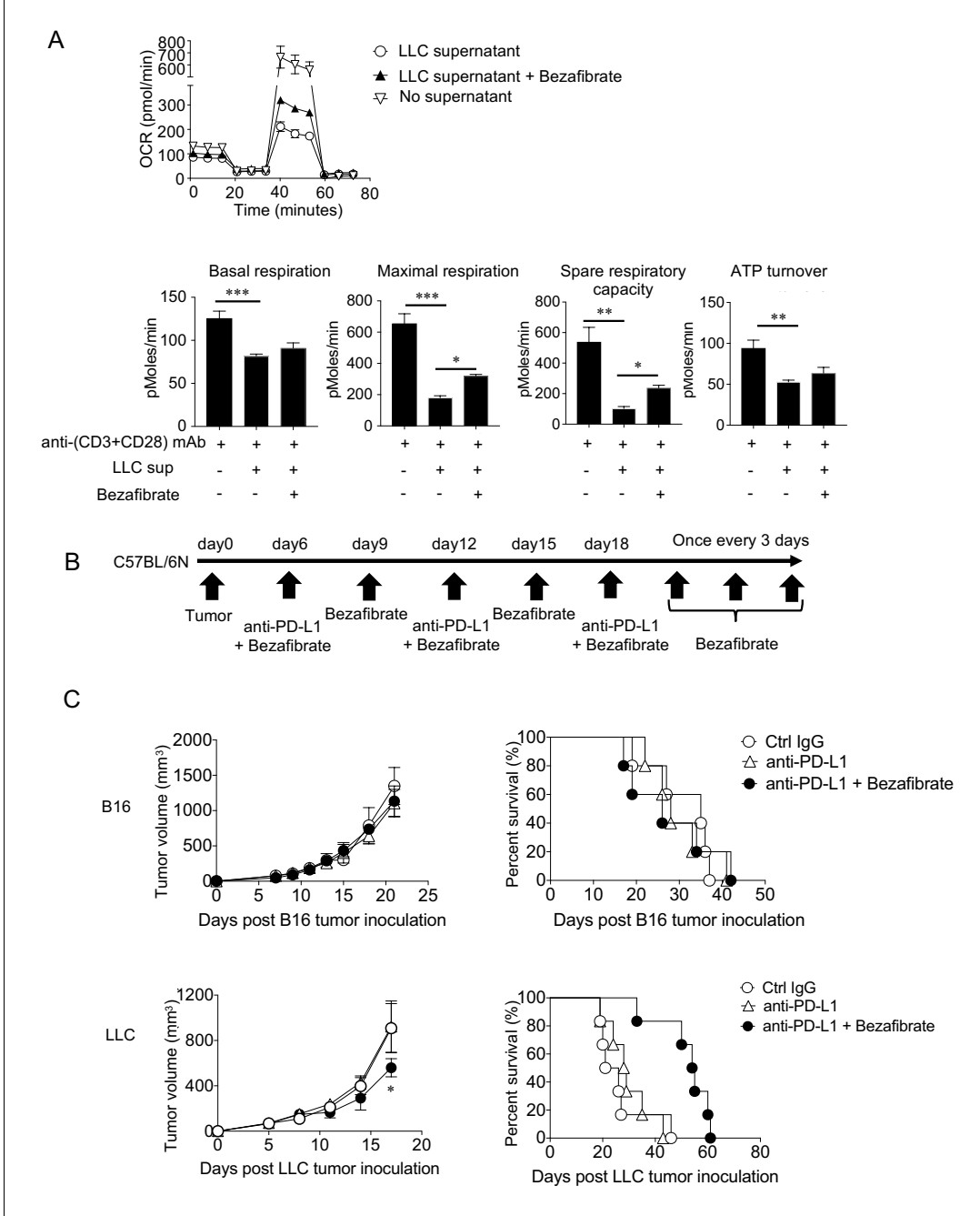

**Figure 7.** Enhancing mitochondrial activation by bezafibrate partially overcomes suppression and improves survival of SIP-positive tumor-bearing hosts in vivo. (**A**) Naïve CD8[+] T cells purified from spleen and LNs of C57BL/6N mice were stimulated for 48 hrs with anti-(CD3+CD28) mAb along with LLC culture supernatant and Bezafibrate (5 μM). Following incubation, OCR of T cells was measured. Data represent the means ± SEM of three wells. *p<0.05, **p<0.01, ***p<0.001, one-way ANOVA analysis. Data are representative of three independent experiments. (**B**) Unresponsive tumors (B16 and LLC) were injected and the mice were treated with anti-PD-L1 mAb along with Bezafibrate (5 mg/kg). Schematic diagram of the combination therapy schedule is shown. (**C**) Tumor graph (left) and survival curve (right) are shown for the B16 (upper panel) and LLC (lower panel) tumor-bearing host treated with Bezafibrate combination therapy. Data represent the means ± SEM of five mice. *p<0.05, one-way ANOVA analysis. Data are representative of three independent experiments.

The online version of this article includes the following figure supplement(s) for figure 7:

**Figure supplement 1.** Enhancing mitochondrial activation by bezafibrate chemicals improves the anti-tumor effect for SIP-positive tumor in BALB/c background.

immunity to some extent, it is reasonable that LLC but not B16 is susceptible to the combination therapy.

Mitochondrial activation is essential for the full activation of T cells. In our in vitro assay system for mitochondrial activities, we stimulated naïve CD8$^+$ T cells by anti-(CD3+CD28) mAb beads because CD28 in addition to CD3 signal is necessary for robust mitochondrial activation during the proliferation (Klein Geltink et al., 2017). Although our OCR data suggest that the suppressive factors downregulate the mitochondrial activity, ECAR also severely inhibited. Therefore, the suppressive factors may inhibit glycolysis, resulting in the attenuation of subsequent OXPHOS reactions. This hypothesis agrees with the fact that T cells rely on glycolysis more than OXPHOS when they differentiate from naïve to effector T cells (Menk et al., 2018). Another possible mechanism for suppression of mitochondrial function by the suppressive factors is inhibition of the downstream signals of CD3 and/or CD28 because these two signals are necessary for the upregulation of glycolysis and OXPHOS in T cells.

In this work, we applied bezafibrate to unresponsive LLC or CT26 tumors. We found this combination therapy partially restored the PD-1 blockade effect per the in vitro assays where bezafibrate partially removed the mitochondrial inhibition by the suppressive factors in the supernatant. This partial effect suggests that under the situation of 'brake' induced by the suppressive factors, the 'acceleration' by PGC-1α/PPAR activation would not fully work. To obtain the maximum benefit, we need to define the suppressive factors and remove the 'brake'. Our data suggest the possibility of unknown small molecules for suppressive factors. The purification of this small molecule by bioassays will enable us to identify its structure by mass spectrometry. Once we know such a compound, we may be able to find the enzymes responsible for the synthesis of this product and target them for combinatorial treatment.

## Materials and methods

### Animals

C57BL/6N and BALB/c inbred mice were purchased from 'The Charles River Laboratories, Japan (Kanagawa, Japan)'. Pdcd1$^{-/-}$ and Rag2$^{-/-}$ inbred mice lines were maintained under specific pathogen-free conditions at the Institute of Laboratory Animals, Graduate School of Medicine, Kyoto University. Female, 6–8 weeks-old mice were used in all the experiments.

### Cell culture

Cell lines were cultured in RPMI or DMEM medium (Gibco, Grand Island, NY, USA; catalog #11875–093 and 11995–065 respectively) with 10% (v/v) heat-inactivated fetal bovine serum and 1% (v/v) penicillin-streptomycin mixed solution (Nacalai Tesque, Kyoto, Japan, 26253–84) as per the instructions recommended by the ATCC. Cell lines were free of mycoplasma contamination. Cell cultures were maintained at 37°C with 5% $CO_2$ in a humidified incubator. Details of different murine cell lines used in the experiment e.g. source of cell lines, background, and origin of cancer, etc. are mentioned in Table 1. The tumor cell lines MethA and GL261 were passaged in vivo once before use in experiments.

### Monotherapy model using anti-PD-L1 antibody

Tumor cells were intradermally (i.d.) injected into the right flank of mice (day 0). Monotherapy with the anti-PD-L1 antibody was started when the tumor size reached 50–60 mm$^3$ (around day 5). Mice were intraperitoneally (i.p.) injected with 80 µg of anti-PD-L1 mAb (clone 1-111A.4); mAb injection was repeated every fifth day. For untreated mice, an isotype control for the anti-PD-L1 mAb (Rat IgG2a, κ) was injected. Tumor sizes were measured every alternate day using a digimatic caliper (Mitutoyo Europe GmbH, Germany) and tumor volume was calculated using the formula for a typical ellipsoid [π × (length ×breadth × height)/6].

### Bilateral tumor model

First, unresponsive tumor cells were i.d.- injected into the left flank of mice (day 0). When the size of the unresponsive tumor was around 60–70 mm$^3$ (around day 6–7), responsive tumor cells were i.d.-injected into the right flank. Two-three days after the responsive tumor injection (around day 9–10),

anti-PD-L1 antibody was injected following a monotherapy treatment model (for the dose of antibody and interval between two injections). Tumor sizes of responsive and unresponsive tumors were measured every alternate day and tumor volume was calculated according to the formula mentioned earlier.

## Chemical reagents

Bezafibrate (Santa Cruz Biotechnology, Dallas, TX, USA) was used at the dose of 5 mg/kg for in vivo combination therapy. Bezafibrate was freshly prepared, immediately before use, in DMSO. Dissolved bezafibrate was diluted in PBS and 200 μL was i.p.-injected per mouse. Bezafibrate was added at the concentration of 5 μM for in vitro assays throughout this work wherever it is used unless specified.

## Combination therapy model

For combination therapy experiments, the therapy started when the tumor size was 60–70 mm$^3$. Mice were i.p.- injected with 40 μg of anti-PD-L1 mAb (clone 1-111A.4); the mAb injection was repeated every sixth day. Mice were i.p.-injected with bezafibrate at 5 mg/kg dose every third day. For control groups, an isotype control for the anti-PD-L1 mAb (Rat IgG2a, κ) and DMSO vehicle for bezafibrate were injected. All groups were subjected to the same dose of DMSO. Tumor measurement was performed as stated above.

## Naïve CD8$^+$ T cell isolation

To isolate naïve CD8$^+$ T cells from C57BL/6N inbred wild-type mice, the spleen and three LNs (axillary, brachial, and inguinal LNs) from both the right and left sides were harvested. The spleen was minced, treated with ACK lysis buffer (0.15 M $NH_4Cl$ + 1.0 mM $KHCO_3$ + 0.1 mM $Na_2$-EDTA) for 2 min to lyse the erythrocytes, and mixed with pooled and minced LN cells. Naïve (CD62L$^{high}$ CD44$^{low}$) CD8$^+$ T cells were then purified from total pooled lymphocytes according to the manufacturer's instructions (Miltenyi Biotec, 130-096-543). For in vitro analysis, naïve CD8$^+$ T cells were stimulated with anti-CD3 and CD28 mAb-coated dynabeads (Thermo Fisher Scientific, Gibco, Catalog# 11452D).

## Collection of culture supernatants from different cell lines

We seeded 0.5 million cells/well in 6-well plates in 4 mL total volume of respective media as recommended by the ATCC. After 48 hrs of incubation, we harvested the culture supernatant, centrifuged at 10,000 x $g$ for 15 min at 22°C, collected the supernatant, and kept it at −80°C for storage. We added culture supernatant one-fourth of the total volume in the well (96-well round-bottom plate) throughout the in vitro assays with naïve CD8$^+$ T cells in this work, unless specified.

## Thymidine incorporation assay

Thymidine solution diluted in spleen RPMI (Basal RPMI media with 10% FCS, 1% Penicillin-Streptomycin, 50 μM 2-Mercapto ethanol, L-Glutamine, Na-pyruvate, NEAA) was added to cells and incubated for 4 hrs at 37°C in a humidified incubator with 5% $CO_2$. After incubation, cells were transferred to a 96-well filter plate followed by the addition of scintillation buffer. Thymidine uptake was measured on a Microbeta$^2$ microplate counter (PerkinElmer, # 2450–0120) machine.

## Heat-inactivation treatment of supernatant

To inactivate the protein component, culture supernatant was boiled for 10 min at 95°C followed by centrifugation at 10,000 x $g$ for 30 min. The supernatant was collected and stored at −80°C for storage.

## Dextran-coated charcoal (DCC) treatment of supernatant

To remove small molecules, the supernatant was treated with DCC, which removes small molecules (e.g. nucleotides, vitamins, lipids) from the sample by adsorbing them on the surface. To remove small molecules, 12 mg DCC (for 500 μL supernatant) was added and incubated for 20 min at 25°C, followed by centrifugation at 10,000 x $g$ for 30 min. After centrifugation, the supernatant that was free from small molecules was collected.

## Fractionation of culture supernatant

Cultures supernatants were fractionated into different fractions using amicon ultra-centrifugal filters (Merck Millipore Ltd., Ireland) with cut-off sizes of 3 KDa and 50 KDa. Supernatants were added to 3 KDa filter and centrifuged at 12,000 x $g$ for 30 min at 4°C. The filtered supernatant was collected and further fractionated using a higher cut-off filter (50 KDa) in a similar way.

## Cell preparation for analysis

For draining lymph node (DLN) analysis, axillary, brachial, and inguinal LNs (one of each) were harvested from the tumor-bearing side (left or right flank) of mice. All LNs were minced and pooled. Average LN cell numbers (total pooled LN cells/3) were used as absolute cell numbers. For tumor-infiltrating lymphocyte (TIL) analysis, tumor tissue was harvested and cut into 1–2 mm pieces with scissors followed by digestion with collagenase type IV (Worthington Biochemical Corporation, Lakewood, NJ, Catalog # LS004188) using a gentle MACS Dissociator (Miltenyi Biotec). The numbers of TILs per mg of tumor tissue were used as the absolute numbers.

## Flow cytometry analysis

The following monoclonal antibodies (mAbs) were used to detect the respective antigens during FACS staining: CD8 (53–6.7), CD62L (MEL-14), CD44 (IM7), CD45.2 (104), T-bet (4B10), IFN-γ (XMG-1.2) from BioLegend (San Diego, CA, USA); and Ki67 (SolA15) from eBioscience (San Diego, CA, USA). All flow cytometry experiments were performed on a FACS Canto II (BD Biosciences, Franklin Lakes, NJ, USA), and analyzed using the FlowJo software (FLOWJO, LLC, Ashland, OR, USA).

Mitochondrial mass, membrane potential, mitochondrial superoxide, and cellular ROS were determined by MitoTracker Green, MitoTracker Deep Red, MitoSOX Red, and CellROX Green reagents, respectively (all from Life Technologies, Carlsbad, CA, USA). The cells were washed twice with D-PBS buffer followed by the addition of dye solution with final concentrations of 0.125, 0.125, 5.0, and 0.625 µM, respectively, in RPMI media and incubated at 37°C in a 5% $CO_2$ humidified incubator for 30 min. After incubation, cells were washed twice with D-PBS buffer followed by surface staining.

## Intranuclear staining

For intranuclear staining, cells were fixed and permeabilized using the Foxp3 staining kit (Thermo Fisher Scientific, Catalog # 00-5523-00) following the manufacturer's instructions. After fixation and permeabilization, cells were incubated with the respective antibody for 15 min at 4°C in the dark, followed by washing with FACS buffer (PBS, 0.5–1% BSA or 5–10% FBS, 0.1% NaN3 sodium azide).

## Intracellular cytokine staining for IFN- γ

Homogenized tumor mass cells from in vivo treated experimental mice were incubated for 4 hrs at 37°C in a 5% $CO_2$ humidified incubator. After incubation, Brefeldin A and Monensin (eBioscience, Invitrogen, Carlsbad, CA, USA; catalog # 4506–51 and 4505–51 respectively) were added at the concentration of 5 µg/mL and 2 µM as per the manufacturer's instructions and incubated for further 2 hrs. Following a total of six hours of incubation, cells were washed once with D-PBS and further stained for surface proteins, if any. Cells were then fixed with 1.5% paraformaldehyde solution (incubated for 15 min at 4°C) and washed twice with FACS buffer. Cells were then treated with 0.5% Triton-X-100 in PBS and incubated for 15 min at 4°C to permeabilize the cells. Monoclonal antibodies to IFN- γ were added (the concentration was pre-optimized) and incubated for 15 min at 4°C followed by washing with FACS buffer.

## qRT-PCR

We isolated RNA from the experimental groups with the RNeasy mini kit (QIAGEN, Hilden, Germany) and synthesized cDNA by reverse transcription (Invitrogen). The primers used to perform quantitative reverse transcription PCR (qRT-PCR) are listed here. The primers pairs used were FP: TACCACCCCATCTGGTCATT, RP: GGACGTTTTGTTTGGTTGGT for CD39; FP: CAAATCCCACACAACCACTG, RP: TGCTCACTTGGTCACAGGAC for CD73; FP: CAAGGGAGTCTGGAACATTG, RP: ACCCAGGTCCTCGCTTATGA for COX2; FP: ATGAGTACACGAAGCCGAGG, RP: CCAGTATTACAGGAGTGACCCAG for mPGES1; FP: CACTGAGCACGGACGGACTGAGA, RP: TCCAATGCTTTCAGGTCTTGACGC for IDO1; FP: CGGAAATCATATCCAACCAG, RP: TGAGGACCGC

TAGCAAGTTTG for PGC-1α; FP: GGTGTTCGGTGAGATTGTAGAG, RP: GTGATAAAACCGTGCTTC
TGG for PGC-1β; and FP: TATTGGCAACGAGCGGTTCC, RP: GGCATAGAGGTCTTTACGGATGT
for β-actin. β-actin was used as loading control.

## Measurement of oxygen consumption rates and extracellular acidification rate

The oxygen consumption rate (OCR) and extracellular acidification rate (ECAR) of treated cells were measured using an XF$^e$96 Extracellular Flux analyzer (Seahorse Biosciences, North Billerica, MA, USA). One day before the experiment, first the XF$^e$96 plate was coated with CellTak solution as per the manufacturer's recommendation. On the day of the experiment, all chemicals (e.g. Oligomycin, FCCP, and Rotenone/Antimycin A) were prepared in OCR media as per the manufacturer's recommendation and the machine was calibrated using the calibrant buffer in the calibrant plate prior to the experiment. 400 thousand cells per well were seeded in the precoated XF$^e$96 plate and the OCR/ECAR was measured. Different parameters from the OCR graph were calculated. ATP turnover was defined as follows: (last rate measurement before oligomycin) - (minimum rate measurement after oligomycin injection). Maximal respiration was defined as follows: (maximum rate measurement after FCCP) - (non-mitochondrial respiration). Spare respiratory capacity (SRC) was calculated by subtracting basal respiration from maximal respiration. We measured the ECAR value in the same well, which contained an optimal glucose level so the basal ECAR (or glycolysis) value is the reading we obtained immediately before oligomycin injection. We prepared the assay medium as described in the XF cell Mito Stress Test Kit (Kit 103015–100). The glucose concentration in this medium is 10 mM. In the classical glycolytic assay procedure (glucose-free media) the final concentration of glucose added to the port was 10 mM while measuring flux. The basal ECAR value in this classical method is calculated by subtracting the last rate measurement before the glucose injection from the maximum rate measurement before the oligomycin injection, which gives essentially the same value if calculated by our method. Glycolytic capacity was defined as the rate measured after the oligomycin injection. The glycolytic reserve was defined as follows: (glycolytic capacity) – (basal ECAR value).

## Statistics

Statistical analysis was performed using Prism 6 (GraphPad Software, La Jolla, CA, USA). One-way ANOVA analysis followed by Sidak's multiple comparison test was utilized to analyze three or more variables. To compare two groups, student's *t*-test was used. All statistical tests were two-sided assuming parametric data, and a *p*-value of <0.05 was considered significant. The variations of data were evaluated as the means ± standard error of the mean (SEM). Five or more samples were thought to be appropriate for the sample size estimate in this study. Samples and animals were randomly chosen from the pool and treated. No blinding test was used for the treatment of samples and animals.

## Study approval

Mice were maintained under specific pathogen-free conditions at the Institute of Laboratory Animals, Graduate School of Medicine, Kyoto University under the direction of the Institutional Review Board.

## Acknowledgements

We thank M Al-Habsi, M Akrami, T Oura, R Hatae, Y Nakajima, R M Menzes, K Yurimoto and Y Kitawaki for assistance in sample preparation.

# Additional information

### Funding

| Funder | Grant reference number | Author |
|---|---|---|
| Japan Agency for Medical Research and Development | JP19gm0710012 | Tasuku Honjo |

| Japan Agency for Medical Research and Development | JP19cm0106302 | Tasuku Honjo |
| Tang Prize | Tang Prize Foundation | Tasuku Honjo |
| Japan Agency for Medical Research and Development | JP191k1403006 | Kenji Chamoto |
| Japan Society for the Promotion of Science | JP16H06149 | Kenji Chamoto |
| Japan Society for the Promotion of Science | 17K19593 | Kenji Chamoto |
| Japan Society for the Promotion of Science | 17F17119 | Partha S Chowdhury |
| Japan Society for the Promotion of Science | 18J15051 | Alok Kumar |
| Cell Science Research Foundation | | Kenji Chamoto |

The funders had no role in study design, data collection and interpretation, or the decision to submit the work for publication.

### Author contributions

Alok Kumar, Conceptualization, Resources, Data curation, Software, Formal analysis, Validation, Investigation, Methodology, Writing - original draft, Writing - review and editing; Kenji Chamoto, Data curation, Methodology, Writing - review and editing; Partha S Chowdhury, Resources, Data curation; Tasuku Honjo, Conceptualization, Supervision, Funding acquisition, Project administration, Writing - review and editing

### Author ORCIDs

Alok Kumar (iD) https://orcid.org/0000-0002-4269-3971
Kenji Chamoto (iD) https://orcid.org/0000-0001-8625-3612
Tasuku Honjo (iD) https://orcid.org/0000-0003-2300-3928

### Ethics

Animal experimentation: His study was performed in strict accordance with the recommendations in the Guide for the Care and Use of Laboratory Animals of the National Institutes of Health. All of the animals were handled according to approved institutional animal care and use committee of Kyoto University. The protocol was approved by the Committee on the Ethics of Animal Experiments of the Kyoto University (Permit Number: Medkyo19080). All surgery was performed under sodium pentobarbital anesthesia, and every effort was made to minimize suffering.

### Decision letter and Author response

Decision letter https://doi.org/10.7554/eLife.52330.sa1
Author response https://doi.org/10.7554/eLife.52330.sa2

## Additional files

### Supplementary files

• Supplementary file 1. List of primers for quantifying mouse gene transcripts by qRT-PCR (related to *Figure 6—figure supplements 3* and *4*).

• Transparent reporting form

### Data availability

PCR primer sequences relating to Figure 6—figure supplements 3 and 4 have been added as Supplementary File 1.

The following datasets were generated:

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
