## [Decision Letter]

Thank you for submitting your article "Tumors attenuating the mitochondrial activity in T cells escape from PD-1 blockade therapy" for consideration by *eLife*. Your article has been reviewed by three peer reviewers, and the evaluation has been overseen by a Guest Reviewing Editor and Jeffrey Settleman as the Senior Editor. The following individual involved in review of your submission has agreed to reveal their identity: Madhusudhanan Sukmar (Reviewer #2).

The reviewers have discussed the reviews with one another and the Reviewing Editor has drafted this decision to help you prepare a revised submission.

Summary:

Kumar A et al. described non-protein (heat stable) small molecule (less than 3kDa) soluble factors from murine tumor cells might be responsible for systemic T cell suppression through impairment of mitochondrial activity in vitro and in vivo by using bilateral transplanted in vivo tumor models and various metabolic and mitochondria assays. The authors classified responsive tumors (MC38 and GL271 in B6 mice, MethA in BALB/c mice) and unresponsive tumors (LLC and Pan02 in B6, CT26 in BALB/c) with this soluble factor mechanism. (B16 is an unresponsive tumor due to MHC loss, not due to soluble factors). The factors also downregulated glycolysis as evidenced by decreased ECAR. The mitochondrial activators bezafibrate partially reversed this immunosuppression in vivo. The authors concluded that such a soluble factor from tumor cells is one of the systemic immunosuppressive mechanisms through impairment of energy metabolism of T cells, and may be responsible for unresponsiveness of some cancers to PD-1 blockade.

Essential revisions:

1) This is an intriguing study showing that soluble factors from tumor cells systemically inhibit energy metabolism of anti-tumor T cells. Although the authors clearly demonstrated presence of non-protein small molecule soluble factors derived from some of the PD-1 Ab unresponsive murine tumor cell lines in multiple tumors and mice, some of the results are duplication of the previous reports such as the authors' own work (bezafibrates and mitochondrial activation potentiating immunotherapy) and others' reports such as T cell suppression by small molecules from tumor cells, and unresponsiveness of B16 through MHC class I loss. Therefore, the responsible molecules may be identified or their unique activity on energy metabolic pathway different from the previously reported immunosuppressive factors should be shown.

2) It is not clear whether the new soluble factors (systemic immunosuppressive property (SIP) factor) really affect specifically on energy metabolism including mitochondrial activity and glycolysis. The factors might non-specifically inhibit T cell activation and proliferation, resulting in impaired energy metabolism. Without identification of the molecules, the authors need to show the unique property of the factors that preferential inhibit energy metabolism pathway, not other T cell activation signaling and pathway, etc. Because the authors clearly showed multiple responsive and unresponsive murine tumor cell lines, comparative studies of metabolome for responsible molecules, and transcriptome for responsible enzymes and pathways, may be performed, and the results will lead to the mechanistic insights for the authors' intriguing findings and lead to the major advance in cancer immunology.

3) It is important to clarify if the impairment of mitochondrial activity by the SIP factors is dependent on PGC-1α and mitochondria biogenesis, because the authors and others (Scharping et al. Immunity 2016) have previously shown that mitochondrial biogenesis is inhibited in T cells, and bezafibrate enhances PGC-1α expression.

4) The authors analyzed T cells only in draining lymph nodes and T cell induction level. It is also important to analyze the effects on effector T cells in tumors. Transcriptional profiling of T cells cultured with boiled LLC supernatant for various times will provide some insights into the nature of the SIP factors.

5) The timing of implant of unresponsive tumors and responsive tumors in in vivo experiments may be changed. For example, responsive tumors on day 0, followed by injection of unresponsive tumors on day 6. The experiments will characterize the nature of the SIP factors.

6) Because the SIP factors inhibited T cells in vitro, similar activity is also be evaluated in various human cancer cell lines.

---

## [Author Response]

Essential revisions:1) This is an intriguing study showing that soluble factors from tumor cells systemically inhibit energy metabolism of anti-tumor T cells. Although the authors clearly demonstrated presence of non-protein small molecule soluble factors derived from some of the PD-1 Ab unresponsive murine tumor cell lines in multiple tumors and mice, some of the results are duplication of the previous reports such as the authors' own work (bezafibrates and mitochondrial activation potentiating immunotherapy) and others' reports such as T cell suppression by small molecules from tumor cells, and unresponsiveness of B16 through MHC class I loss. Therefore, the responsible molecules may be identified or their unique activity on energy metabolic pathway different from the previously reported immunosuppressive factors should be shown.

Bezafibrate (Chowdhury et al., 2018) and other mitochondrial activation chemicals (Chamoto et al., 2017) have been used in the previous paper to potentiate immunotherapy on responsive tumor (MC38 and MethA). In this paper, we have worked on the unresponsive mechanism. We first showed that bezafibrate in combination with PD-L1 mAb treatment improves the survival of the mice with unresponsive tumor (LLC and CT26), which produce T cell mitochondria-suppressive factors.

We tested the possibility of small molecules, which were previously reported (e.g., adenosine, PGE2 and kynurenine) to inhibit T cell proliferation. We quantified tumor cells transcript levels of responsible enzyme for above small molecules (e.g., surface enzymes CD39 and CD73 for generating adenosine and AMP; COX2 and mPGES1 for prostaglandin E2; IDO1 for generating suppressive factors after tryptophan degradation). However, we could not find a quantitative correlation between SIP-positive property and transcripts associated with above factor (please see Figure 6—figure supplement 4). Texts are updated in the manuscript (subsection “Secretion of immune inhibitory small molecules from SIP-positive tumors”, last paragraph).

Regarding ‘B16 through MHC class I loss’, we think these data are necessary due to the following two reasons: 1) it is known that B16 has different sublines that vary in their phenotypic expression according to laboratories. We need to specify the phenotype and unresponsive mechanism of our B16. 2) Contrast between B16 and LLC is important to confirm that LLC employ systemic immunosuppressive mechanism, but not local ignorance. We added this point in the subsection “The immunotherapy-resistant B16 tumor employs local immunological ignorance”.

2) It is not clear whether the new soluble factors (systemic immunosuppressive property (SIP) factor) really affect specifically on energy metabolism including mitochondrial activity and glycolysis. The factors might non-specifically inhibit T cell activation and proliferation, resulting in impaired energy metabolism. Without identification of the molecules, the authors need to show the unique property of the factors that preferential inhibit energy metabolism pathway, not other T cell activation signaling and pathway, etc. Because the authors clearly showed multiple responsive and unresponsive murine tumor cell lines, comparative studies of metabolome for responsible molecules, and transcriptome for responsible enzymes and pathways, may be performed, and the results will lead to the mechanistic insights for the authors' intriguing findings and lead to the major advance in cancer immunology.

To answer the reviewers’ question whether SIP factor directly inhibit T cell mitochondria or not, naïve CD8^+^ T cells were cocultured with B16 (SIP-negative) or LLC (SIP-positive) supernatant only for 2 hours and mitochondrial activation parameters were studied. We found mitochondrial mass, potential, superoxide and cellular ROS were downregulated in the presence of LLC supernatant within 2 hours. This result indicates that LLC soluble suppressive factors inhibit mitochondria directly, but not mediated by cellular transcriptional activity. Further, to examine whether the mitochondrial inhibition by LLC SIP factor is specific to CD8^+^ T cells or non-specifically inhibit mitochondria, we performed the same experiment on purified B cells (B220^+^ cells). We found the mitochondrial phenotype of B cells was reduced when cocultured with LLC supernatants but not with B16 supernatants. Therefore, LLC SIP factors inhibit mitochondria directly in general although they would affect less on cancer cells as they rely majorly upon glycolysis for energy demand. We have added these results in Figure 6F and Figure 6—figure supplement 3 and updated the manuscript text accordingly (subsection “Secretion of immune inhibitory small molecules from SIP-positive tumors”, second paragraph; Figure 6F legend).

3) It is important to clarify if the impairment of mitochondrial activity by the SIP factors is dependent on PGC-1α and mitochondria biogenesis, because the authors and others (Scharping et al. Immunity 2016) have previously shown that mitochondrial biogenesis is inhibited in T cells, and bezafibrate enhances PGC-1α expression.

Since suppressive factors are downregulating the mitochondria in 2 hours, it is unlikely that PGC-1α protein expression or other transcription factors are involved. Indeed, the mRNA levels of PGC-1α/β in CD8^+^ T cells did not change in 2 hours when cocultured with supernatant as shown in Figure 6—figure supplement 3A. We updated the text in the revised manuscript accordingly (subsection “Secretion of immune inhibitory small molecules from SIP-positive tumors”, second paragraph).

4) The authors analyzed T cells only in draining lymph nodes and T cell induction level. It is also important to analyze the effects on effector T cells in tumors. Transcriptional profiling of T cells cultured with boiled LLC supernatant for various times will provide some insights into the nature of the SIP factors.

We analyzed T cells in tumor as well as draining lymph node. As shown in Figure 1E-F, Figure 2B-C, effector functions of intra-tumor T cells (T-bet and IFN-γ) and mitochondria activation parameters were higher in PD-1 blockade group compared to ctrl IgG group in responsive tumor-bearing hosts. As mentioned above in our response to comment #2, the suppressive factor inhibits mitochondria directly. It is unlikely that there will be drastic change in transcriptome in such a short time as 2 hours. Indeed, the transcript level of PGC-1α did not change as we have mentioned before.

5) The timing of implant of unresponsive tumors and responsive tumors in in vivo experiments may be changed. For example, responsive tumors on day 0, followed by injection of unresponsive tumors on day 6. The experiments will characterize the nature of the SIP factors.

We think swapping of tumors would not provide further insight in understanding the mechanism of unresponsiveness. Our purpose is to understand the immunosuppression caused by unresponsive tumors on the host’s immune response. We have experienced that when the responsive tumor got big size on day 8-9, PD-1 blockade effect became weak (data not shown). In such situation, it is difficult to address the properties of SIP factors.

6) Because the SIP factors inhibited T cells in vitro, similar activity is also be evaluated in various human cancer cell lines.

As per reviewers’ advice, we performed in vitro proliferation assay using the supernatants from human cancer cells. We found supernatants from some human cancer cell lines (like Colo205, ES2) inhibits T cells proliferation while some do not (like DU145, A549). We have included these data as ‘Figure 6—figure supplement 2’ and updated in the revised manuscript at relevant places (subsection “Secretion of immune inhibitory small molecules from SIP-positive tumors”, first paragraph).